# Prediction of Agro-Morphological and Nutritional Traits in Ethiopian Mustard Leaves (*Brassica Carinata* A. Braun) by Visible-Near-Infrared Spectroscopy

**DOI:** 10.3390/foods8010006

**Published:** 2018-12-22

**Authors:** Damián Martínez-Valdivieso, Rafael Font, Mercedes Del Río-Celestino

**Affiliations:** 1Department of Genomics and Biotecnology, IFAPA Center La Mojonera, Camino San Nicolás 1, La Mojonera, 04745 Almería, Spain; damianvaldivieso@gmail.com; 2Department of Food Science and Health, IFAPA Center La Mojonera, Camino San Nicolás 1, La Mojonera, 04745 Almería, Spain; rafaelm.font@juntadeandalucia.com

**Keywords:** *Brassicaceae*, near-infrared spectroscopy (NIRS), flowering, pubescence leaf, total phenolic content

## Abstract

The particular characteristics of some of the Ethiopian mustard accessions available from seed banks could be used to increase the production and the diversity of products available to consumers and to improve their general quality. The objectives of this study were to determine the genetic variability for agro-morphological (days to first flowering: DFF and leaf pubescence: LP) and nutritional traits (total phenolic content: TPC) among accessions, and to evaluate the potential of near-infrared spectroscopy (NIRS) to predict these traits in Ethiopian mustard leaves. A great variation was found for the traits evaluated. The reference values were regressed against different spectral transformations by modified partial least-squares (MPLS) regression. The coefficients of determination in cross-validation (R^2^cv) shown by the equations for DFF, LP and TPC were 0.95, 0.63 and 0.99, respectively. The standard deviation to standard error of cross-validation ratio (RPD), were for these traits, as follows: DFF: 4.52, LP: 1.53 and, TPC: 24.50. These results show that the equations developed for DFF and TPC in Ethiopian mustard, can be predicted with sufficient accuracy for screening purposes and quality control, respectively. In addition, the LP equation can be used to identify those samples with “low”, “medium” and “high” groups. From the study of the mean and deviation standard spectra, and regression vectors of MPLS models it can be concluded that some major cell components, highly participated in modelling the equations for these traits.

## 1. Introduction

Changes in production patterns in agriculture and dietary habits are important factors in the increasing attention to neglected species used on their own or mixed in leaf salads. This is the case of several species from the *Brassicaceae* family, such as Ethiopian mustard (*Brassica carinata* A. Braun), whose cultivation is thought to have started about 4000 years B.C. in the Ethiopian highlands. One of the reasons why the *Brassica* species have increased in popularity is because they deliver high concentrations of health-promoting bioactive phytochemicals, such as glucosinolates and phenolic compounds.

Previous works have revealed a large degree of variability in Ethiopian mustard for agro-morphological characters, such as flowering time, plant height, leaf number/plant, leaf bloom and leaf blade blistering, seed yield and in main seed storage components [1,2]. For its use as leafy vegetables (4th generation of vegetables), the following traits are preferred: Low leaf pubescence, late flowering, many leaves per plant, tolerance to major diseases and pests and high concentration of compounds possessing antioxidant capacity [2,3]. Breeding programmes are being conducted to encourage Ethiopian mustard cultivation and consumption at La Mojonera (Almería, Spain). In this sense, efforts are being taken to characterize agro-morphological (leaf pubescence and number of days to flowering) and antioxidant traits (such as phenolic compounds) in *B. carinata* lines especially from local populations. 

There is evidence that leaf pubescence is a structural adaptation against insect and fungal attack, for heavy metal detoxification purposes, but also to enhance tolerance to low and high temperature [4,5,6,7,8]. 

The day to first flowering is an agronomical trait which is recommended to be as high as possible, due to leaves are harvested before shooting and flowering. Leaves can be harvested after 20–30 days and then sequentially harvested from regrowth. Sequential planting combined with repeated harvest may be a way to manage this crop despite high bolting under high temperatures [9].

In relation to the total phenolic acid content previous works have shown variability for the total phenolic compounds in *Brassica juncea* leaves with values between 6.51 to 14.9 mg/g DM (dry matter) [10,11], although higher content have also been found in other Cruciferous species, such as rocket (4474.5 to 32,700 μg g^−1^ dw) demonstrating that are an excellent source of these antioxidant compounds [4]. 

The high cost and labour input required for characterizing flowering time, leaf pubescence and total phenolic content by liquid chromatography or Ultraviolet-visible Spectrophotometry in Ethiopian mustard leaves are serious handicaps to analyse large sets of samples, which is usually necessary to identify the target genotypes in screening programs.

In contrast, the use of fast analytical techniques, such as Near-Infrared Spectroscopy (NIRS) results in many advantages, since analysis can be carried out with a considerable saving of time, at a low cost and without using hazardous chemicals [12]. Many authors have used this technique for determining quantitative analysis in environmental, agricultural and food research in *Brassica* spp [13,14,15]. To our knowledge, no studies have been reported on the use of the NIRS technique for predicting the days to first flowering (DFF), leaf pubescence (LP) and total phenolic content (TPC) in Ethiopian mustard leaves. Our objectives were: (1) to determine the genetic variability for DFF, LP and TPC among accessions, and (2) to evaluate the potential of NIRS to predict these traits in Ethiopian mustard leaves. In addition, we provide some knowledge about the mechanism used by NIRS for determining these characters successfully in the leaves of this species. 

## 2. Results and Discussion

### 2.1. Reference Analysis of Days to First Flowering, Leaf Pubescence and Total Phenolic Content in the Accessions

In the conservation of plant genetic resources, the availability of characterization data and information on diversity help germplasm users identify the accessions of interest and also provide plant breeders initial data for use in crop improvement programs. The results of the present study (Figure 1) revealed a high phenotypic variability in the accessions of Ethiopian mustard studied. Concerning the DFF trait, it ranged from 77 to 137 days, with a mean value of 108 DFF. As the plants are harvested before shooting and flowering, DFF is recommended to be as high as possible. Some authors have also reported DFF values in Ethiopian mustard, similar or lower than those shown in this work (108 DFF). Thus, comparable DFF values were reported by De Haro et al. [1] (range: 95–126, mean: 111 DFF) for plants grown in Spain. Plants grown in dry regions flower earlier and produce ripe seeds within four months from sowing in comparison with vegetable crops grown with adequate moisture which produce seeds in 5–6 months [16]. Other authors indicated lower DFF values, Muthoni [2] reported 82 DFF (ranging between 64–96 DFF,) while Mnzava and Schippers [16] indicated differences between oil and vegetable types of Ethiopian mustard; thus, oil types start flowering about 70 days, while the vegetable cultivars after about 84 days, depending on cultivar and growing conditions.

Regarding LP, the accessions exhibited a great variation for this trait from glabrous to trichome abundance (Figure 1). Tcacenco et al. [17] also reported a large variation for leaf pubescence in this species.

Among phytochemicals possessing antioxidant capacity, phenolic compounds are one of the most important groups. The total phenolic content for the leaves of the Ethiopian mustard accessions showed large variation (Figure 1) with mean values of 8.57 mg/g dry matter (DM) and varying from 2.20 to 12.70 mg/g DM. The TPC of this species has been less studied with regard to other species of *Brassica*, although it was similar to previously reported results in *Brassica juncea* with mean values of 7.95 mg/g DM [10,11].

### 2.2. Spectral Data Pre-Treatments and Equation Performances

#### 2.2.1. Second Derivative Spectra of Ethiopian Mustard Leaf

Table 1 shows the mean concentrations, standard deviations, and ranges of the different traits analyzed by the reference methods. It can be seen from Table 1 that the traits under study showed a wide range of variation.

Figure 2 shows the mean spectrum without preprocessing (Figure 2a), the mean spectra (Figure 2b) and Standard Deviation spectra (Figure 2c) using Standard normal variate and detrend transformations and second derivative treatment (2, 5, 5, 2)) of the freeze-dried Ethiopian mustard samples used to conduct this work (*n* = 135). The average spectrum without (Figure 2a) and with preprocessing (Figure 2b) showed absorption bands in the visible region of the spectrum with a maximum at λ = 676 nm, which corresponds to electronic transitions in the red, which has been assigned to absorption by chlorophyll [18].

The NIR region of the spectrum (Figure 2b) showed characteristic absorption bands at 1432 and 1916 nm related to O–H stretch second and first overtones of water, respectively; 1726, 2310 and 2348 nm related to C–H stretch first overtones and combination bands of lipids [19]; at 2058 nm related to N–H stretch of amides [20] and 2000 and 2274 nm related to O–H + C–O deformation, O–H stretch plus deformation, and O–H + C–C stretch of starch respectively [20]. The highest SDs of the spectral data (Figure 2c) were found in the visible region of the spectrum (electronic transitions in the red) and also at 1434 nm, due to the first overtone of O–H stretching; at 1694 nm related to S–H stretch first overtone or C–H stretch first overtone of CH_3_ groups; at 1728 and 1764 nm related to C–H stretching by methylene groups; 1800 to 2000 nm, related to S–H stretch first; 1980 to 2174 assigned to N–H stretch of amides; 2270 nm which has been assigned to O–H plus C–C stretch groups of cellulose; and 2300, influenced to C–H combination tones [10,20].

#### 2.2.2. Calibration Equation

Following the considerations reported by Shenk and Westerhaus [21] about the estimation of the accuracy of a calibration equation from cross-validation, the R^2^_CV_ obtained for DFF, LP and TPC were indicative of equations with good quantitative information. For these traits, the mathematical models developed explained from 63 to 99% of the variance contained in the data (Table 1). 

For a comparison of the potential of the prediction among the equations obtained, a standardization of the different SECVs is needed. In this way, the RPD ratio (The standard deviation (SD) to standard error of cross-validation ratio (SECV)) [10] was estimated for each equation. RPDs in Ethiopian mustard were as follows: DFF: 4.5, LP: 1.52 and, TPC: 24.73. In general terms RPDs reported in this work in Ethiopian mustard are higher for TPC (RPD: 24.73) than those obtained for *Brassica napus* seed (1.20) and tea leaves [22] and lower than those attained for rice grain (RPD: 47.1) [23].

From the data reported in this work it is concluded that DFF and TPC in Ethiopian mustard can be predicted with sufficient accuracy for screening purposes and quality control, respectively, while the LP equation can be used for a correct separation of the samples into “low”, “medium” and “high” groups.

#### 2.2.3. Modified Partial Least Square Loadings

Previous work has reported NIRS predictions for the flowering cycle of *Brassica juncea* plants through the glucosinolate analysis of the seed [24]. These authors postulated that genotypes containing mainly sinigrin showed a larger flowering cycle. Sinigrin is the main glucosinolate in Ethiopian mustard seeds and leaves [25]. Although it is difficult to state with precision the degree of participation of the major cell constituents of leaves (protein, lipids, starch, cellulose, etc.) in the calibration of DFF, on the basis of the similarities between the MPLS (modified partial least-squares) loading for the DFF trait (Figure 3a) and the spectrum of the commercial sinigrin (Figure 4) previously reported [13], it seems that the molecule of sinigrin has a specific signal in the leaf spectrum of Ethiopian mustard (bands at 1694, 1980, 2172 and 2310, 2348 nm).

Concerning LP calibration, in previous work for desert species weaker correlation was found between this trait and NIR (R^2^cv < 0.25) than those obtained in this study [26]. Prediction of this trait has to be done on the basis of secondary correlations with plant components. This phenomenon is supported by data from regression vectors of MPLS model for LP (Figure 3b).In the visible region of the spectrum, chromophores (absorption at 672 nm) also participated actively in constructing the third term. Previous studies evidence that many plants accumulate heavy metals, but also UV-absorbing compounds, such as flavonols in trichomes or leaf pubescence which further protect the photosynthetic tissues from damaging amount of radiations (UV-A and UV-B) and were used to develop satisfactory NIRS predictions [4,7]. The third term was influenced by absorption bands (1399–1699 nm) characteristic to vibrations of the C–H and O–H bonds, corresponding to water and phenolic absorbance [27,28,29]. It can also be concluded from Figure 3b, where the group N–H of amides (1980 and 2172 nm); and the C–H groups of oil and fibre (2348 nm) strongly influenced the third MPLS loadings of the equation for this trait. 

Regarding TPC calibration, in the Figure 3c from the third regression vector of MPLS model, characteristic bands for phenolics can be observed in the regions from 1415 nm and 1512 nm and from 1955 to 2035 nm as previously has been reported by Zhang et al. [23].

## 3. Materials and Methods 

### 3.1. Plant Material and Greenhouse Experiments 

Forty-five *B. carinata* accessions originating from Ethiopia were sown at the IFAPA Center in La Mojonera (Almería, Spain). The accessions were randomized and planted in three replicates by direct seeding in 7-m, single-row plots located at 36°47’19” N, 02°42’ 11” W. 

Twenty plants at the five-leaf stage in each one of the 45 accessions per replication were used to study agro-morphological traits (LP and DFF).

### 3.2. Determination of LP

Leaf pubescence or trichomes was determined according to the International Board for Plant Genetic Resources according to the descriptors for *Brassica* and *Raphanus* [30] where 0—glabrous, 1—very sparse, 3—sparse, 5—intermediate, 7—abundant.

### 3.3. Determination of DFF

The number of days from seed sowing to the appearance of the first open flower was taken for determining the DFF. 

### 3.4. Determination of the Total Phenolic Fraction

For TPC analysis three individual plants from each one of the 45 accessions per replication were collected at five weeks after sowing. For each plant, 3 or 4 leaves were washed with tap water, weighed to assess their biomass, and placed in Ziploc-type freezer bags at −20 °C for post-harvest storage. The samples were freeze-dried up to performing the TPC analysis.

The concentration of total phenolic compounds (TPC) was estimated by a modified version of the Folin–Ciocalteu method [31], using gallic acid as standard, for which a calibration curve was run with solutions of 50, 100, 200, 300, 400, 500 and 600 mg/L of this compound. A 0.06 mL aliquot of extract 1.58 mL of distilled water, 0.1 mL of Folin–Ciocalteu reagent and 0.3 mL of Na_2_CO_3_ (20% *w*/*v*) were mixed and heated at 50 °C for 5 min. After 30 min, the absorbance was measured at 765 nm against a blank similarly prepared, but containing 70:30 ethanol–water mixture (pH 3.2) instead of extract. Sodium carbonate (Panreac), Folin–Ciocalteu reagent (FCR) and gallic acid (both from Sigma–Aldrich) were used to determine the total phenol fraction. The absorbance was measured with a ThermoSpectronic UV–visible Spectrometer (Thermo Fisher Scientific, Waltham, MA, USA).

### 3.5. NIRS Analysis

All the samples which were previously analysed by reference method were then analysed in a NIRS monochromator (NIR Systems mod. 6500, NIR Systems, Inc., Silversprings, MD, USA). In this work, each spectrum was recorded in triplicate from each sample, and was obtained as an average of 32 scans over the sample, plus 16 scans over the standard ceramic. Samples were placed in a small ring cup (3.75 cm φ) using the spinning sample module and their spectra collected between 400–2500 nm, registering the absorbance values (log 1/R) at 2 nm intervals for each sample.

Prediction equations for DFF, LP and TPC were developed using the ISI program CALIBRATE (WINISI II, Infrasoft International, LLC, Port Matilda, PA, USA) with the modified partial least squares regression option. Modified partial least squares (PLSm) was used as a regression method to correlate the spectral information (raw optical data or derived spectra) of the samples and TPC and TCC contents determined by the reference method, using a different number of wavelengths from 400 to 2500 nm for the calculation. The objective was to perform a linear regression in a new coordinate system with a lower dimensionality than the original space of the independent variables. The PLS loading factors (latent variables) were determined by the maximum variance of the independent (spectral data) variables and by a maximum correlation with the dependent (chemical) variables. The model obtained used only the most important factors, the “noise” being encapsulated in the less important factors.

Cross-validation was performed on the calibration set to determine both, the ability to predict unknown samples and the best number of terms to use in the equation [21]. Cross-validation is an internal validation method that like the external validation approach seeks to validate the calibration model on independent test data, but it does not waste data for testing only, as occurs in external validation. This procedure is useful because all available chemical analyses for all individuals can be used to determine the calibration model without the need to maintain separate validation and calibration sets. The method is carried out by splitting the calibration set into M segments and then calibrating M times, each time testing about a (1/M) part of the calibration set.

Calibration equations were computed using four mathematical treatments (0, 0, 1, 1 (derivative, gap, smooth, second smooth); 1, 4, 4, 1; 1, 10, 10, 1 and 2, 5, 5, 1) on the calibration set. Standard normal variate and detrend transformations were used to correct scattering, and two passes were the option chosen to eliminate outliers (spectra with a standardized distance from the mean (H) > 3 (Mahalonobis distance)), by using principal component analysis (PCA). The objective of this procedure was to detect and, if necessary, remove possible samples whose spectra differed from the other spectra in the set.

Wavelengths from 400 to 2500 nm every 2 nm, were used for calibration. The standard error of calibration (SEC), coefficient of determination (R^2^cv), standard error of cross-validation (SECV) and 1-VR (1 minus the ratio of unexplained variance to total variance) statistics were used to characterise the different equations obtained and to determine the best calibration equation [32]. SECV was used as an estimate of the standard error of performance (SEP) [33]. The best calibration equations were obtained with the 2,5,5,1 mathematical treatments (second derivative of the raw optical data, with a gap of 5 nm and 5 and 2 nm for the first and second smooth) for all traits evaluated. The regression vectors of the three factors generated from the MPLS method performed on the 2,5,5,1 mathematical treatments were calculated. The loading plots show the regression coefficients of each wavelength to the parameter being calibrated for each factor of the equation.

## 4. Conclusions

The results of this study indicate that there is a wide variation in Ethiopian mustard collection held at IFAPA La Mojonera based on vegetative agro-morphological and nutritional traits and therefore it offers opportunities for selection and breeding. In addition, the results obtained showed that NIRS can be used as a screening technique to evaluate a large number of samples in a short period of time. Once the screening process is done, reference methods can be used to accurately determine these parameters in samples previously selected by NIRS, avoiding the need of the analyses.

## Figures and Tables

**Figure 1 foods-08-00006-f001:**
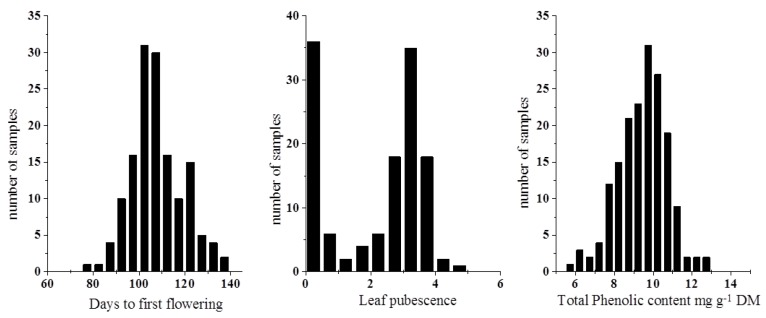
Distribution plots for DFF, LP and TCP in the whole set of samples (*n* = 135 for DFF and LP, and *n* = 405 for TPC). DFF, days to first flowering; LP, leaf pubescence; TPC, total phenolic content; DM, dry matter.

**Figure 2 foods-08-00006-f002:**
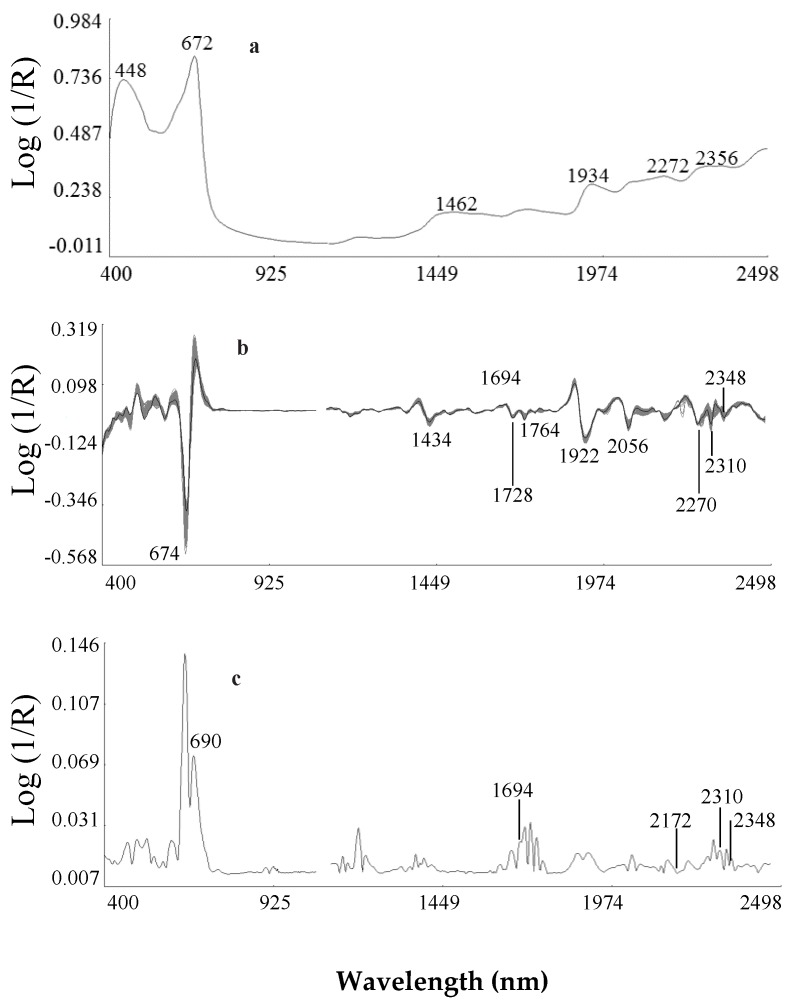
Near-infrared mean spectrum without preprocessing (**a**); Near-infrared mean spectrum using Standard normal variate and detrend transformations and second derivative (**b**) and standard deviation using Standard normal variate and detrend transformations and second derivative (**c**) of Ethiopian mustard samples. (R: is reflectance)

**Figure 3 foods-08-00006-f003:**
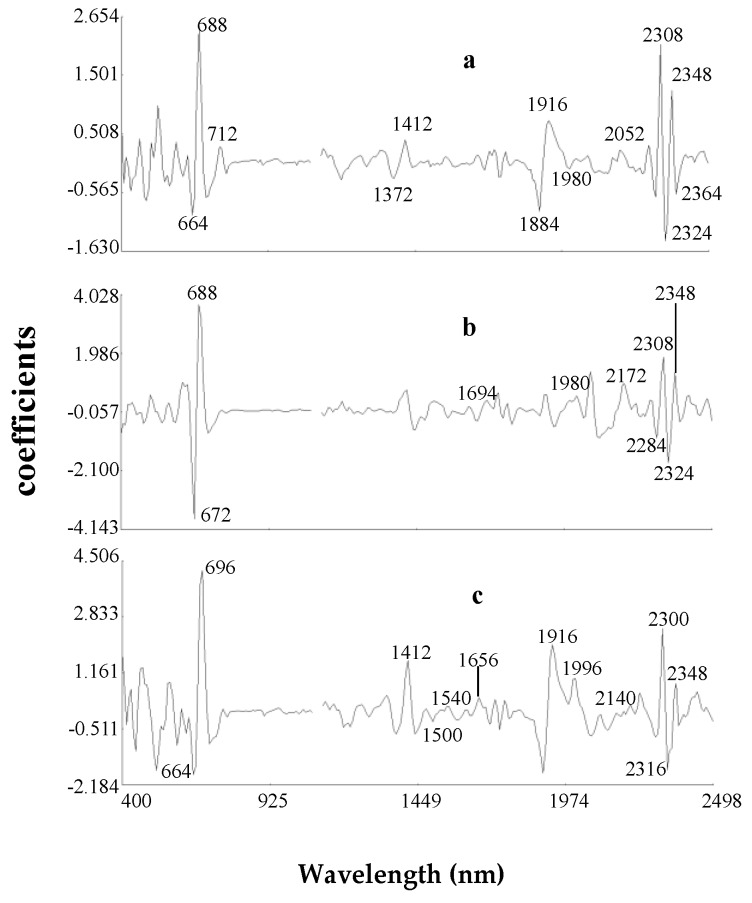
The third regression vector of MPLS (modified partial least-squares) models for DFF (**a**), LP (**b**) and TPC (**c**) in Ethiopian mustard leaves in the second derivative (2, 5, 5, 2; Standard normal variate and detrend transformations).

**Figure 4 foods-08-00006-f004:**
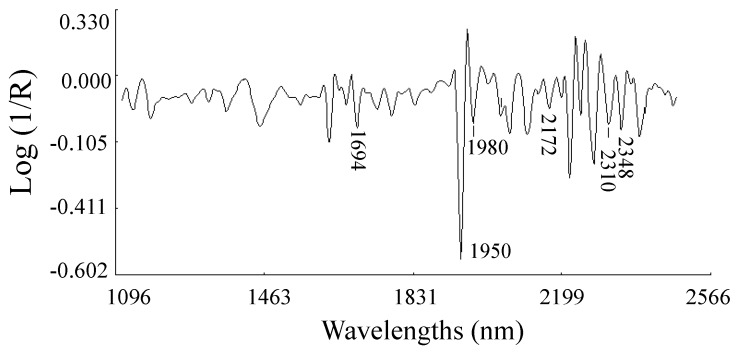
Near-infrared mean spectrum (2, 5, 5, 2; Standard normal variate (SNV) + detrend transformations (DT)) of commercial sinigrin.

**Table 1 foods-08-00006-t001:** Calibration and cross-validation statistics for the different equations developed for agro-morphological and nutritional traits in Ethiopian mustard leaves.

	Calibration						Cross-Validation		
Traits	*n*	Range	Mean	SD ^d^	SEC ^e^	R^2 f^	SECV ^g^	R^2^cv ^h^	RPD ^i^
**DFF ^a^**	135	82–137	107.53	10.87	1.43	0.98	2.40	0.95	4.52
**LP ^b^**	135	0–7	3.42	0.80	0.32	0.84	0.52	0.63	1.53
**TPC ^c^**	405	2.20–12.70	8.57	1.96	0.06	0.99	0.08	0.99	24.50

^a^ days to first flowering; ^b^ leaf pubescence; ^c^ total phenolic content; ^d^ SD: standard deviation of the reference data; ^e^ SEC: standard error of calibration; ^f^ R^2^: coefficient of determination of the calibration; ^g^ SECV: standard error of cross-validation; ^h^ R^2^cv: coefficient of determination of the cross-validation; ^i^ RPD: ratio SD to SECV. DFF, days to first flowering; LP, leaf pubescence; TPC, total phenolic content; DM, dry matter.

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
