# Peer review of "Prediction of Agro-Morphological and Nutritional Traits in Ethiopian Mustard Leaves (Brassica Carinata A. Braun) by Visible-Near-Infrared Spectroscopy"

_foods, 2018, doi:10.3390/foods8010006_

Round 1

Reviewer 1 Report

The results of the paper are of significance and relevance to the journal. The use of NIRS on predicting agro-morphological traits is novel. There are only a few minor comments found below:

When comparing results to other published work should mention how results are similar or different and just not mention this in the text. For example, lines 85-86 of page 3 a range of values are compared for DFF between results presented and those of literature. It is better to compare means as comparing ranges is broad. Another example is lines 92-94 of page 3 where TPC is compared. Results are shown to be 2.2-12.70 mg/g and compare with 6.51-14.9 mg/g. It is mentioned that results are within range of previous work. Should mention in what sense. Are values between 2.2-6.51 mg/g for samples not that significant? Again should mention and compare mean values of results when comparing results with previous work.

A very minor change is to label x axes for Figure 2 and 3 and elsewhere. For example use wavelength (nm).

Author Response

Answer to the referee #1:

Answer to the referee #1:

Authors thank the reviewer for their critical evaluation and relevant comments of the manuscript. They have been addressed as follow:

The results of the paper are of significance and relevance to the journal. The use of NIRS on predicting agro-morphological traits is novel. There are only a few minor comments found below:

1.     When comparing results to other published work should mention how results are similar or different and just not mention this in the text. For example, lines 85-86 of page 3 a range of values are compared for DFF between results presented and those of literature. It is better to compare means as comparing ranges is broad. Another example is lines 92-94 of page 3 where TPC is compared. Results are shown to be 2.2-12.70 mg/g and compare with 6.51-14.9 mg/g. It is mentioned that results are within range of previous work. Should mention in what sense. Are values between 2.2-6.51 mg/g for samples not that significant? Again should mention and compare mean values of results when comparing results with previous work.

Authors agree with the referee´s comments and we have compared mean values for both, DFF (lines 98-108), and TPC (lines 114-117)

2.     very minor change is to label x axes for Figure 2 and 3 and elsewhere. For example use wavelength (nm).

Following the referee’s recommendationthe labels have been included for  Figures 2 (line 165) and 3 (line 201)

Reviewer 2 Report

Introduction

Page 1, line 17
It says LF, but later it is refered to as LP

Results and discussion

Page 2, line 77
The subsection says carotenoids but I cannot find anywhere else in the text where this is discussed?

Page 3, line 101
”It can be seen…” I am not sure what that sentence means. Could the author be more specific how they evaluate that the data is ”good for all the criteria under instestigation”.

Page 4, line 111
I suggest that authors also show the raw data without preprocessing. For figure 2 it is confusing that the y-axis is labelled 2nd Derivative.

Page 5, line 136
0.63 to 0.99%. Maybe the authors mean 63-99%?

Page 5, line 155-156
I suggest the authors include the spectrum of sinigrin

Page 6
Figure 3. The author call these for loadings, but I would say these are the regression vectors for each model?

Materials and Methods

Page 7-8, line 218-232
I suggest provide more information on the program ISI CALIBRATE. Who is the vendor?

I suggest that the authors provide information on the number of acquired spectra/scans on each sample

Why did the authors choose MPLS over PLS? Might be a good idea to give a reference so the reader can see what the modification is.

I suggest the authour explain what passes means and how this is used to remove outliers.

How was the cross validation made? Leave-one-sample-out? Leave-one-replicate-out? etc. The calibration equations/models does not make much sense as long as the reader is not providede with the information on how these models were cross validated.

Author Response

Answer to the referee #2:

The authors would like to thank the reviewer for their valuable comments and suggestions to improve the quality of the manuscript.

Page 1, line 17
It says LF, but later it is refered to as LP

The LF acronym has been changed by LP (Line 18)

Results and discussion

Page 2, line 77
The subsection says carotenoids but I cannot find anywhere else in the text where this is discussed?

Thanks to the referee for this point. The subsection has been changed (line 88-89)

Page 3, line 101
”It can be seen…” I am not sure what that sentence means. Could the author be more specific how they evaluate that the data is ”good for all the criteria under investigation”.

The sentence has been revised to clarify (lines 132-133)

Page 4, line 111
I suggest that authors also show the raw data without preprocessing. For figure 2 it is confusing that the y-axis is labelled 2nd Derivative.

Authors agree with the referee´s comments and we have included the raw data (line  165) and we have changed the label by log (1/R) (line 165)

Page 5, line 136                          0.63 to 0.99%. Maybe the authors mean 63-99%?

Thanks to the referee for this point. The change has been done (line 174)

Page 5, line 155-156
I suggest the authors include the spectrum of sinigrin

The spectrum of sinigrin has been included (line 224)

 Page 6
Figure 3. The author call these for loadings, but I would say these are the regression vectors for each model?

The legend has been changed (line 202)

Materials and Methods

Page 7-8, line 218-232. I suggest provide more information on the program ISI CALIBRATE. Who is the vendor?

Following the referee’s recommendationthe vendor of the program ISI CALIBRATE has been included (line 270)

I suggest that the authors provide information on the number of acquired spectra/scans on each sample

The information has been included (line 264-266)

Why did the authors choose MPLS over PLS? Might be a good idea to give a reference so the reader can see what the modification is.

The justification of why the MPLS regression was used has been included in the manuscript (lines 271-279)

I suggest the author explain what passes means and how this is used to remove outliers.

Following the referee’s recommendation more information about the identification and removal of outliers has been included in the manuscript (lines 301-304)

How was the cross validation made? Leave-one-sample-out?Leave-one-replicate-out?etc. The calibration equations/models does not make much sense as long as the reader is not providede with the information on how these models were cross validated.

The information about how the cross validation was made has been included in the manuscript (lines 280-297)

Reviewer 3 Report

Near infrared spectroscopy has been gaining momentum as a non-destructive method in agriculture testing. Current manuscript presents a very convincing case of the same. There is a huge potential in NIR in agriculture produce quality control. However, there are few things that need to be explained in order to make this manuscript publishable.

1)The experiment is very well designed, however, the writing of the manuscript is very hard to follow. The sentences are long and there are minor grammatical mistakes.

2)One of my major comment is about the leaf pubescence phenomenon. Do you think using green house controlled environment resulted in weak regressions with the NIR data?

Author Response

Answer to the referee #3:

I am grateful to the reviewer for the time dedicated to the revision of the manuscript. In the revised version of the paper, we carefully took into account the suggestions passed for the amelioration of the manuscript.

Near infrared spectroscopy has been gaining momentum as a non-destructive method in agriculture testing. Current manuscript presents a very convincing case of the same. There is a huge potential in NIR in agriculture produce quality control. However, there are few things that need to be explained in order to make this manuscript publishable.

The experiment is very well designed, however, the writing of the manuscript is very hard to follow. The sentences are long and there are minor grammatical mistakes.

Following the referee’s recommendation the manuscript has been revised by an English speaker.

One of my major comment is about the leaf pubescence phenomenon. Do you think using green house controlled environment resulted in weak regressions with the NIR data?

The trichome density in Ethiopian mustard leaf is controlled by the genotype of the plant, specifically it  is governed by two genes. Therefore, if the plants were grown under the same controlled environment, the differences found among accessions were due to genotype. I believe that greenhouse controlled environment should not affect the regression model. The cause of this weak regression could be attributed to the fact that the prediction of this trait has to be done on the basis of secondary correlations
